# Real-World Concordance between Germline and Tumour *BRCA1/2* Status in Epithelial Ovarian Cancer

**DOI:** 10.3390/cancers16010177

**Published:** 2023-12-29

**Authors:** Robert D. Morgan, George J. Burghel, Helene Schlecht, Andrew R. Clamp, Jurjees Hasan, Claire L. Mitchell, Zena Salih, Joseph Shaw, Sudha Desai, Gordon C. Jayson, Emma R. Woodward, D. Gareth R. Evans

**Affiliations:** 1Department of Medical Oncology, The Christie NHS Foundation Trust, Wilmslow Road, Manchester M20 4BX, UK; 2Division of Cancer Sciences, Faculty of Biology, Medicine and Health, School of Medical Sciences, University of Manchester, Manchester M13 9PL, UK; 3North West Genomic Laboratory Hub, Manchester University NHS Foundation Trust, Oxford Road, Manchester M13 9WL, UK; 4Division of Evolution, Infection and Genomics, Faculty of Biology, Medicine and Health, School of Biological Sciences, University of Manchester, Manchester M13 9PL, UK; 5Department of Gynaecological Pathology, Manchester University NHS Foundation Trust, Oxford Road, Manchester M13 9WL, UK; 6Department of Pathology, The Christie NHS Foundation Trust, Wilmslow Road, Manchester M20 4BX, UK; 7Department of Clinical Genetics, Manchester University NHS Foundation Trust, Oxford Road, Manchester M13 9WL, UK

**Keywords:** ovarian cancer, germline, somatic, *BRCA1*, *BRCA2*

## Abstract

**Simple Summary:**

Approximately 10–15% of patients with epithelial ovarian cancer have an inherited (germline) *BRCA1* or *BRCA2* mutation. Following a diagnosis of epithelial ovarian cancer, patients are routinely tested for germline and/or somatic (tumour) *BRCA1*/*2* mutations. Our study shows that if germline *BRCA1/2* testing is only performed for patients with a positive tumour *BRCA1/2* test result, and tumour testing is performed using Myriad’s myChoice^®^ companion diagnostic, a proportion of germline *BRCA1/2* large rearrangements could be missed. If paired germline-tumour DNA testing is not possible for all patients, our data shows that it would be appropriate to test all patients with epithelial ovarian cancer aged < 79 years old for germline *BRCA1/2* mutations, regardless of the tumour *BRCA1/2* result, whilst only needing to test patients aged ≥ 80 years old for a germline *BRCA1/2* mutation if they have a positive tumour *BRCA1/2* result.

**Abstract:**

Patients diagnosed with epithelial ovarian cancer may undergo reflex tumour *BRCA1*/*2* testing followed by germline *BRCA1/2* testing in patients with a positive tumour test result. This testing model relies on tumour *BRCA1*/*2* tests being able to detect all types of pathogenic variant. We analysed germline and tumour *BRCA1/2* test results from patients treated for epithelial ovarian cancer at our specialist oncological referral centre. Tumour *BRCA1/2* testing was performed using the next-generation sequencing (NGS)-based myChoice^®^ companion diagnostic (CDx; Myriad Genetics, Inc.). Germline *BRCA1/2* testing was performed in the North West Genomic Laboratory Hub using NGS and multiplex ligation-dependent probe amplification. Between 11 April 2021 and 11 October 2023, 382 patients were successfully tested for tumour *BRCA1* and *BRCA2* variants. Of these, 367 (96.1%) patients were tested for germline *BRCA1*/*2* variants. In those patients who underwent tumour and germline testing, 15.3% (56/367) had a *BRCA1*/*2* pathogenic variant (36 germline and 20 somatic). All germline *BRCA1/2* pathogenic small sequencing variants were detected in tumour DNA. By contrast, 3 out of 8 germline *BRCA1/2* pathogenic large rearrangements were not reported in tumour DNA. The overall concordance of germline *BRCA1/2* pathogenic variants detected in germline and tumour DNA was clinically acceptable at 91.7% (33/36). The myChoice^®^ CDx was able to detect most germline *BRCA1/2* pathogenic variants in tumour DNA, although a proportion of pathogenic large rearrangements were not reported. If Myriad’s myChoice^®^ CDx is used for tumour *BRCA1/2* testing, our data supports a testing strategy of germline and tumour *BRCA1/2* testing in all patients diagnosed with epithelial ovarian cancer aged < 79 years old, with germline *BRCA1/2* testing only necessary for patients aged ≥ 80 years old with a tumour *BRCA1/2* pathogenic variant.

## 1. Introduction

Ovarian cancer is the most lethal gynaecological cancer, with more than 200,000 deaths attributed to the disease each year worldwide [1]. The high mortality rate associated with ovarian cancer occurs largely because, at the time of diagnosis, ovarian cancer cells have metastasized to the peritoneum, meaning curing it is highly unlikely [2]. Indeed, the five-year overall survival for advanced-stage epithelial ovarian cancer is poor at around 30% to 35% [3]. Significant improvements in our understanding of the biology of epithelial ovarian cancer has led to new targeted therapies and subsequent improvement in survival outcomes. This is best exemplified by the synthetic lethal use of poly (ADP-ribose) polymerase-1/2 inhibitors (PARPi) to treat high-grade serous carcinomas that harbour a *BRCA1* or *BRCA2* pathogenic/likely pathogenic variant [4]. Registration trials for maintenance PARPi therapies in newly diagnosed or relapsed platinum-sensitive *BRCA1/2*-mutant high-grade serous or endometrioid carcinoma have reported some of the most impressive hazard ratios for survival outcomes seen in oncology [5,6,7,8,9,10,11].

To identify patients eligible for first line PARPi maintenance therapy, reflex tumour testing for *BRCA1/2* pathogenic/likely pathogenic variants has been implemented for all patients diagnosed with advanced-stage high-grade epithelial ovarian cancer [12]. Subsequent germline *BRCA1/2* testing is then requested for patients with a positive tumour *BRCA1/2* test result, thereby determining whether the pathogenic/likely pathogenic variant is constitutional (i.e., germline) or somatically acquired (i.e., tumour-only) [13]. Those patients with a germline *BRCA1/2* pathogenic/likely pathogenic variant are referred to clinical genetics to access familial cancer services. This model of selectively testing only certain index cases (i.e., those with a positive tumour test result) for germline *BRCA1/2* pathogenic variants relies on tumour tests being able to detect all types of *BRCA1/2* pathogenic/likely pathogenic variant, including point mutations, small insertions and deletions, and large rearrangements. If a germline *BRCA1/2* pathogenic/likely pathogenic variant is missed, index cases may miss out on maintenance PARPi therapies, as well as breast cancer risk-reduction measures [14]. Moreover, family members will not gain access to cascade testing and breast/ovarian cancer risk-reduction strategies in related, unaffected germline *BRCA1/2* heterozygotes.

There is limited real-world data reporting the number of germline *BRCA1/2* pathogenic/likely pathogenic variants missed by testing tumour DNA using commercially available clinically validated genetic assays [15,16,17,18,19]. The aim of this study was to report the concordance of *BRCA1/2* pathogenic/likely pathogenic variants detected in germline and tumour DNA in a large, real-world cohort of patients treated for epithelial ovarian cancer at our specialist oncological referral centre. Patients underwent local germline *BRCA1/2* testing, whilst tumour testing was performed using the myChoice^®^ companion diagnostic (CDx; Myriad Genetics, Inc., Salt Lake City, UT, USA).

## 2. Materials and Methods

Eligibility criteria included all patients diagnosed with epithelial ovarian, fallopian tube or primary peritoneal (Müllerian type) cancer that were treated at The Christie Hospital (Manchester, UK), and tested for tumour *BRCA1/2* pathogenic/likely pathogenic variants using the myChoice^®^ CDx (Myriad Genetics, Inc., Salt Lake City, UT, USA). Index cases of ovarian carcinosarcoma were eligible for inclusion. All patients diagnosed with non-epithelial ovarian cancer were excluded. All FIGO (International Federation of Gynaecology and Obstetrics) stages of disease were eligible for inclusion. Only patients aged ≥18 years at diagnosis were included.

MyChoice^®^ CDx testing was coordinated by the North West Genomic Laboratory Hub (Manchester, UK) and was available from 11 April 2021 onwards. Tumour *BRCA1/2* testing was planned for all newly diagnosed patients in line with the national genomic test directory for cancer, specified by NHS England [20]. The methodology used in Myriad’s myChoice^®^ CDx has been reported [21]. The myChoice^®^ CDx is a next-generation-sequencing (NGS) in vitro diagnostic assay that reports the homologous recombination deficiency (HRD) status of formalin-fixed paraffin-embedded (FFPE) tumour tissue based on the tumour *BRCA1*/*2* status and the genomic instability score (GIS) [22]. In this study, we focussed on the tumour *BRCA1/2* status only. The myChoice^®^ CDx detects single nucleotide variants, insertions and deletions, and large rearrangements in the protein coding regions and intron/exon boundaries of *BRCA1* and *BRCA2* [22].

Archival FFPE tumour blocks were sent for HRD testing. The percentage tumour cell count was determined locally by a Histopathologist with a specialty interest in gynaecological pathology. Formalin-fixed paraffin-embedded tissue was assessed for tumour content by assessing the total cellularity and tumour cell content, expressed as a percentage of all viable nucleated cells on a haematoxylin and eosin stained 5 μm thick section.

Germline *BRCA1/2* testing was planned for all eligible patients as per the national genomic test directory for rare and inherited disease, specified by NHS England [23]. Germline testing took place locally in the North West Genomic Laboratory Hub (Manchester, UK) using DNA extracted from peripheral lymphocytes. The chemagic Prime™ instrument (Revvity, Inc., Waltham, MA, USA) was used to extract DNA. Enrichment was performed using SureSelect custom designed probes (Agilent Technologies, Inc., Santa Clara, CA, USA) that targeted the coding regions of transcripts (NM_007294.3 [BRCA1] and NM_000059.3 [BRCA2]) including the immediate splice sites +/− 15 base pairs and known intronic pathogenic/likely pathogenic variants in *BRCA1* and *BRCA2*. Next-generation sequencing was performed using the NextSeq 550 System (Illumina, Inc., San Diego, CA, USA). Target coverage was ≥90% of *BRCA1* and *BRCA2* at a depth of at least 100×. Only pathogenic [class 5] and likely pathogenic [class 4] variants were reported (hereon described together as ‘pathogenic variants’) [24,25]. A local, in-house bioinformatic pipeline reported single nucleotide variants, and small insertions, deletions, and duplications < 40 base pairs in length. Only germline pathogenic variants with a variant allele frequency of ≥5% were reported.

Testing for germline large rearrangements in *BRCA1* and *BRCA2* was performed using multiplex ligation-dependent probe amplification (MLPA^®^) [26]. The MLPA^®^ probe kits P002-D1 (BRCA1) and P045-D1 (BRCA2; MRC Holland, Amsterdam, Netherlands) were used to analyse germline DNA. Amplified ligation products were subject to fragment analysis using an ABI 3130xl Genetic Analyser and sized using GeneMapper Software version 6.0 (Thermo Fisher Scientific, Inc., Waltham, MA, USA). Copy number status calling was performed using data exported from GeneMapper using custom-developed spreadsheets that report relative dosage quotient for each probe compared with five reference control samples. All MLPA^®^ tests were performed in duplicate for confirmation of results.

Categorical data were reported as number (percentage). Continuous data were reported as median (range or interquartile range [IQR]). The Mann–Whitney U test was used to measure differences in the median of two groups with statistical significance defined as a *p* value < 0.05.

## 3. Results

### 3.1. Study Group

Between 11 April 2021 and 11 October 2023, 442 tumour samples from 412 patients were tested using Myriad’s myChoice^®^ CDx. The tumour cell content ranged from 5% to 80% (tumour cell content not provided in 61/442 [13.8%] samples). The tumour *BRCA1/2* assay failure rate was 10.2% (45/442 tumours). Those samples that failed tumour testing had a tumour cell content of 5% to 65% (tumour cell content not provided in 10/45 samples). Two or more tumour samples were tested from 29 patients due to prior complete assay failure (tumour *BRCA1/2* and GIS testing; 23/29 patients) or because the patient was mistakenly tested twice (6/29 patients). Tumour *BRCA1/2* testing was successful in 382 patients. Of these, 15 patients (3.9%) did not undergo germline *BRCA1/2* testing. The study group comprised 367 patients who successfully underwent both tumour and germline *BRCA1/2* testing (Table 1). Of these, most patients had been diagnosed with advanced-stage, high-grade, non-mucinous epithelial ovarian cancer.

### 3.2. BRCA1/2 Pathogenic Variants

There were 58 *BRCA1/2* pathogenic variants detected in 56 patients (prevalence 15.3%; 56/367) (Table 2). Thirty-six patients had a germline *BRCA1/2* pathogenic variant (18 *BRCA1*, 18 *BRCA2*; prevalence 9.8%), and 20 patients had a somatic *BRCA1/2* pathogenic variant (11 *BRCA1*, 9 *BRCA2*; prevalence 5.4%). Two patients had two different *BRCA1* or *BRCA2* pathogenic variants (all somatic). Neither of the samples that harboured two somatic *BRCA1/2* pathogenic variants had a reversion mutation (*BRCA2*:c.3337G>T and *BRCA2*:c.3342_3348del; *BRCA1*:c.457_486delinsC and *BRCA1*:c.524_526delinsTG). There were 51/58 unique *BRCA1/2* pathogenic variants (32 germline, 19 somatic). Of the seven *BRCA1/2* pathogenic variants detected twice, five were *BRCA2* (four germline, one somatic). The somatic pathogenic variant detected twice was *BRCA2*:c.4688G>A.

The median age at diagnosis in patients with a germline or somatic *BRCA1/2* pathogenic variant was 58 years old (IQR: 51–69 years old) and 69 years old (IQR: 57–77 years old), respectively (*p* = 0.018). *BRCA1/2* pathogenic variants were more frequently detected in patients diagnosed with high-grade serous carcinoma (54 versus 2 [non-high-grade serous carcinoma]) and those with advanced-stage disease (8 [FIGO stage I/II] versus 48 [FIGO stage III/IV]).

Most *BRCA1/2* pathogenic variants were small sequencing variants (46/58 [79.3%]; 28 germline, 18 somatic) and measured between 1 and 52 base pairs in length. Most small sequencing variants were less than five base pairs long (44/46 [95.7%]). Twelve *BRCA1/2* pathogenic variants were large rearrangements (12/58 [20.7%]; eight germline, four somatic).

### 3.3. Real-World Concordance between Germline and Tumour BRCA1/2 Pathogenic Variants

All patients with a *BRCA1/2* pathogenic variant detected in tumour DNA underwent germline *BRCA1/2* testing. Of the 36 germline *BRCA1/2* pathogenic variants, 33 were detected in tumour DNA. The concordance of germline *BRCA1/2* pathogenic variants in germline and tumour DNA was 91.7% (33/36) (Table 3). All small sequencing *BRCA1/2* pathogenic variants were detected in germline and tumour DNA. The three germline *BRCA1/2* pathogenic variants not detected in tumour DNA were all large rearrangements and included two *BRCA1* Exon 13 duplications (*n* = 2) and a *BRCA2* Exon 1–24 deletion (*n* = 1; Table 4). All germline *BRCA1/2* large rearrangements were detected in tumour samples from patients diagnosed with advanced-stage, high-grade serous carcinoma, aged < 79 years (age range: 43 to 77 years old).

## 4. Discussion

The implementation of mainstream *BRCA1/2* testing for patients diagnosed with epithelial ovarian cancer occurred due to the development of PARPi as standard therapy. Tumour and germline *BRCA1/2* testing can be performed in parallel (i.e., together) or sequentially (i.e., germline-first then tumour, or tumour-first then germline) [27,28]. In the parallel model of testing, germline (often blood) and tumour DNA are tested for all patients, regardless of the result of each test. In the sequential model of testing, only patients with germline *BRCA1/2* wild-type undergo subsequent tumour *BRCA1/2* testing to detect additional somatic *BRCA1/2* pathogenic variants (germline-first), or alternatively, only patients with a tumour *BRCA1/2* pathogenic variant undergo subsequent germline testing to determine whether the variant is constitutional or somatically acquired (tumour-first). The choice of using a parallel versus sequential model of testing is determined by the local treating multi-disciplinary team and is often based on financial costs (including reimbursement agreements) and availability of regional diagnostic services [12]. The national genomic test directory for NHS England allows all patients diagnosed with epithelial ovarian cancer to undergo germline *BRCA1/2* testing, and all patients eligible for first-line treatment, who have been diagnosed with high-grade ovarian cancer, to undergo tumour *BRCA1/2* testing (as part of ‘tumour HRD testing’) [20,22].

Following validation of Myriad’s myChoice^®^ CDx in the randomised, Phase 3 trial, PAOLA-1 reporting that patients with HRD-positive tumours had longer median progression-free survival and overall survival with first-line bevacizumab and olaparib therapy [10,29]; this commercial assay has been adopted by many cancer centres to detect tumour *BRCA1/2* pathogenic variants [30,31]. Our study shows that for cancer centres that use a tumour-first *BRCA1/2* testing approach using the myChoice^®^ CDx, around 5–10% of germline *BRCA1/2* pathogenic variants may not be reported, all of which are likely to be pathogenic large rearrangements. Our real-world data show a higher rate of unreported germline *BRCA1/2* pathogenic variants compared to myChoice^®^ CDx test data from breast and ovarian tumours tested in the randomised, Phase 3 PARPi trials OlympiAD and SOLO2, where the missed rate was much lower at 0.7% and 1.7%, respectively [15,16]. All the missed pathogenic variants in OlympiAD and SOLO2 were pathogenic large rearrangements [15,16]. The higher miss rate in our study is probably due to one of the unreported large rearrangements (*BRCA1* Exon 13 duplication) being a founder mutation in the North of England, from where our cohort is derived [32]. Moreover, the size of our study population (*n* = 56) was lower than that tested in OlympiAD (*n* = 143) and SOLO2 (*n* = 241) [15,16].

It is unclear why both *BRCA1* Exon 13 duplications and a *BRCA2* 14–16 deletion were unreported in our study. The three FFPE samples for these patients contained more than 30% tumour content, preferred by Myriad for testing. Moreover, two out of three of the FFPE samples had a GIS reported, suggesting that the quality and quantity of tumour DNA extracted reached the required threshold for full assay completion (the GIS for the two tumours with a *BRCA1* Exon 13 duplication were 68 and 70). In addition, a germline *BRCA2* Exon 14–16 deletion was reported in tumour DNA from another patient in our study, confirming the ability of the myChoice^®^ CDx to detect this pathogenic large rearrangement. Unfortunately, we were unable to perform repeat HRD testing of other tumour samples from the three patients with these unreported pathogenic variants because the positive germline mutant status had already been reported.

One possible explanation why *BRCA1* Exon 13 duplication was not reported in our study is that this large rearrangement may often exist outside the limit of detection of the myChoice^®^ CDx. In the Phase 3 trial, SOLO2, Myriad reported that both patients with a germline *BRCA1* Exon 13 duplication were not reported by the myChoice^®^ CDx because “it was not possible to confirm that these large rearrangements were intragenic” [15]. The wider implication of possible under reporting of *BRCA1* Exon 13 duplication can be seen in two recent publications in which only one case of this large rearrangement was reported in over 22,000 ovarian cancer tissue samples tested using the myChoice^®^ CDx [33,34]. This unexpectedly low prevalence of *BRCA1* Exon 13 duplication contradicts previous prevalence data from North American/European cohorts of patients at risk of hereditary breast and/or ovarian cancer, where this large rearrangement was the fourth most common germline *BRCA1* pathogenic variant [35,36,37]. Our group is particularly interested in *BRCA1* Exon 13 duplication because it accounts for 3.1% (85/2709) of all germline *BRCA1/2* pathogenic variants in our regional dataset (as of 11 October 2023). Cancer centres located in other geographical regions where germline *BRCA1* Exon 13 duplication occurs frequently should be aware of the possible under reporting of this large rearrangement by the myChoice^®^ CDx [32]. It may also be useful to repeat our study in a different population where *BRCA1* Exon 13 duplications are less common and/or in a population with alternative large *BRCA1/2* founder duplications.

Matched tumour-normal *BRCA1/2* testing can be costly and logistically challenging [12,38]. However, the implications of missed germline pathogenic variants are significant, especially for unaffected family members in which risk-reduction strategies can prevent breast and/or ovarian cancer from occurring [39]. To limit costs of genetic testing, we have proposed a testing strategy that involves paired germline and tumour *BRCA1/2* testing in all patients diagnosed with epithelial ovarian cancer aged < 79 years old, while reserving tumour-first testing for those aged ≥ 80 years old [40]. Our latest study further supports this testing strategy, should the myChoice^®^ CDx be used. The only two *BRCA1/2* pathogenic variants detected in patients diagnosed at ≥80 years old were small sequencing variants that were detected in tumour DNA (*BRCA1*:c.68_69del and *BRCA2*:c.4478_4481del). No large rearrangements occurred in any patient aged > 77 years old. By adopting our age-defined *BRCA1/2* testing strategy, we estimate a 15% reduction in the total number of patients being tested, while maintaining near universal detection of all *BRCA1/2* pathogenic variants.

The main limitations with this study are the modest sample size and single-centre status. We also acknowledge the fact that almost 4% (15/382) of the study group did not undergo germline *BRCA1/2* testing. A review of the clinical notes of these patients showed that almost all either declined germline testing or had poor prognostic clinical factors and died before seeing an oncologist to consent for germline testing. We also recognize that a minority (12/367) of the study group did not have a diagnosis of high-grade ovarian cancer and were therefore ineligible for tumour HRD testing. Only one patient with non-high-grade ovarian cancer had a tumour *BRCA1/2* pathogenic variant. The inclusion of these patients provides a real-world perspective of genetic testing practice at a large tertiary referral centre. It is also important to note that no tumour *BRCA1/2* test has been validated to distinguish between germline and somatic pathogenic variants, so all patients with a tumour *BRCA1/2* pathogenic variant, irrespective of the tumour test used, should undergo germline testing.

## 5. Conclusions

We report the concordance between germline and tumour *BRCA1/2* pathogenic variants in a large, real-world cohort of patients diagnosed with epithelial ovarian cancer. If the Myriad myChoice^®^ CDx is used for tumour HRD testing, our data support a testing model of germline and tumour *BRCA1/2* testing in all patients diagnosed with epithelial ovarian cancer aged < 79 years old, with germline *BRCA1/2* testing only necessary for patients aged ≥ 80 years old with a tumour *BRCA1/2* pathogenic variant.

## Figures and Tables

**Table 1 cancers-16-00177-t001:** Demographic data for study group.

Characteristic	Study Group (367 Patients)
**Age at diagnosis—years**	
Median	66
Range	24–90
**Histology—number (%)**	
High-grade serous	326 (89)
Low-grade serous	7 (2)
Endometrioid *	10 (3)
Clear cell	14 (4)
Mucinous	0
Mixed ^‡^	1 (<1)
Carcinosarcoma	8 (2)
Adenocarcinoma, NOS	1 (<1)
**FIGO stage—number (%)**	
I	10 (3)
II	11 (3)
III	243 (66)
IV	103 (28)

Key: FIGO, International Federation of Gynecology and Obstetrics (2014) staging system; NOS, not otherwise specified. * Five patients diagnosed with grade 1 or 2 (low-grade) endometrioid ovarian adenocarcinoma and five patients diagnosed with grade 3 (high-grade) endometrioid ovarian adenocarcinoma. ^‡^ One patient was diagnosed with clear cell and grade 1 (low-grade) endometrioid ovarian adenocarcinoma.

**Table 2 cancers-16-00177-t002:** Types of *BRCA1/2* pathogenic variants.

Variant Type	Germline *BRCA1/2* (36 Variants)	Somatic *BRCA1/2* (22 Variants)
**Nucleotide level—number (%) ***		
Single nucleotide variants	9 (25)	6 (27)
Small deletions	13 (36)	7 (32)
Small insertions	1 (3)	0
Small duplications	5 (14)	3 (14)
Small indels	0	2 (9)
Large rearrangements	8 (22)	4 (18)
**Protein level—number (%)**		
Nonsense	7 (19)	5 (14)
Frameshift	19 (53)	11 (31)
Missense	1 (3)	0
Unknown ^‡^	9 (25)	6 (17)

Key: Indels, insertion-deletions; * The denominator is the number of variants (36 germline, 22 somatic); ^‡^ Includes splice site variants and large rearrangements.

**Table 3 cancers-16-00177-t003:** Germline and tumour *BRCA1/2* pathogenic variants.

		Germline *BRCA1/2* Status	
		Mutant	Wild-Type
**Tumour *BRCA1/2* status**	Mutant	33	20 *
	Wild-type	3 ^‡^	311

Key: Data presented as number of patients; * Somatic *BRCA1/2* pathogenic variants; ^‡^ False negative germline *BRCA1/2* result.

**Table 4 cancers-16-00177-t004:** Pathogenic large rearrangements detected in germline and tumour DNA.

Large Rearrangement	Germline DNA—Number of Pathogenic Variants Detected	Tumour DNA—Number of Pathogenic Variants Detected
** *BRCA1* **		
Exon 1–2 deletion	1	1
Exon 3 deletion	0	1 *
Exon 9–12 deletion	1	1
Exon 13–24 deletion	0	1 *
Exon 17 deletion	1	1
Exon 20 deletion	1	1
Exon 13 duplication	2	0 ^‡^
** *BRCA2* **		
Exon 1–24 deletion	0	1 *
Exon 1–27 deletion	0	1 *
Exon 14–16 deletion	2	1 ^‡^

Key: * Somatic *BRCA1/2* pathogenic variants; ^‡^ Discordant germline and tumour *BRCA1/2* results.

## Data Availability

The data presented in this study are available on request from the corresponding authors.

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
