# Peer review of "Real-World Concordance between Germline and Tumour BRCA1/2 Status in Epithelial Ovarian Cancer"

_cancers, 2023, doi:10.3390/cancers16010177_

Round 1
Reviewer 1 Report
Comments and Suggestions for Authors
This is a very well written manuscript exploring real world concordance between germline and somatic BRCA mutations using a commercially available somatic testing diagnostic that is widely utilized. The authors demonstrate good concordance between the two tests (aprox 92%) with three germline large deletions missed by somatic testing. Unfortunately the authors were unable to repeat testing on additional samples from these three cases, and acknowledge this as a limitation. Additional limitations include relatively small sample size and single institution. Once again these limitations are acknowledged by the authors.
Minor suggestions to manuscript.
1. Would include strategy to offer pair germline/somatic testing to those <80 y in the abstract
2. Would state that neither sample harboring two mutations had a reversion mutation (slightly different wording to make this clear).
3. Would suggest in MS that it would be useful to repeat this study in a population where E13 deletions are less common.
I feel this is an excellent manuscript/study that adds real world data to this field.
Author Response
Thank you for your review. Our responses are outlined below.
1. Would include strategy to offer pair germline/somatic testing to those <80 y in the abstract.
We have included this in the abstract now.
2. Would state that neither sample harbouring two mutations had a reversion mutation (slightly different wording to make this clear).
We have amended this sentence in the results section now.
3. Would suggest in MS that it would be useful to repeat this study in a population where E13 deletions are less common.
We have included this in the discussion now.
Reviewer 2 Report
Comments and Suggestions for Authors
Overall, this is a very meaningful study. But it would be better if there are improvements in the following aspects:
1.Please provide more background information.
2.In Table 2. Types of BRCA1/2 pathogenicvariants. Please explain more about "Nonsense;Frameshift;Missense."
Author Response
Thank you for your review. Our responses are outlined below.
1. Please provide more background information.
Thank you for your suggestion. We feel the amount of information in the background is appropriate for the study, but would be happy to add further information if Reviewer 2 can suggest the information he/she feels needs adding.
2. In Table 2. Types of BRCA1/2 pathogenic variants. Please explain more about "Nonsense;Frameshift;Missense."
Thank you for your suggestion. These terms are universally used in genetics to describe the type of changes that occur at a protein level (e.g., nonsense and frameshift mutations are truncating mutations that lead to loss-of-function). We do not feel that the terms require further explanation because they are already widely used in genetics, oncology and medical practice.
Reviewer 3 Report
Comments and Suggestions for Authors
I read with great interest the manuscript, which falls within the aim of this Journal and offers a high-quality overview of the topic.
Methodology is accurate and conclusions are supported by the data analysis. The tables are clear and interesting.
Although the manuscript can be considered already of high quality, I would suggest taking into account the following minor recommendations:
- I find it interesting to include a reference to the screening programs for early cervical cancer diagnosis (Golia D'Augè T, Giannini A, Bogani G, Di Dio C, Laganà AS, Di Donato V, Salerno MG, Caserta D, Chiantera V, Vizza E, Muzii L, D’Oria O. Prevention, Screening, Treatment and Follow-Up of Gynecological Cancers: State of Art and Future Perspectives. Clin. Exp. Obstet. Gynecol. 2023, 50(8), 160. https://doi.org/10.31083/j.ceog5008160).
- Inclusion/exclusion criteria should be better clarified by extending their description.
- What are the implications of these findings for clinical practice and/or further research?
It is important to report the results obtained by the authors in the context of clinical practice and to adequately highlight what contribution this study adds to the literature already existing on the topic and to future study perspectives.
Author Response
Thank you for your review. Our responses are below.
1. I find it interesting to include a reference to the screening programs for early cervical cancer diagnosis (Golia D'Augè T, Giannini A, Bogani G, Di Dio C, Laganà AS, Di Donato V, Salerno MG, Caserta D, Chiantera V, Vizza E, Muzii L, D’Oria O. Prevention, Screening, Treatment and Follow-Up of Gynecological Cancers: State of Art and Future Perspectives. Clin. Exp. Obstet. Gynecol. 2023, 50(8), 160. https://doi.org/10.31083/j.ceog5008160).
Thank you for your suggestion. We are unsure of the rationale for including this reference about cervical cancer in our manuscript. Although this is an important paper for cervical cancer, we do not feel the addition of this reference adds to our manuscript.
2. Inclusion/exclusion criteria should be better clarified by extending their description.
Thank you for your suggestion. The inclusion/exclusion criteria was defined at the start of the study. The complete inclusion/exclusion criteria is included in the manuscript. There are no other inclusion/exclusion criteria.
3. What are the implications of these findings for clinical practice and/or further research? It is important to report the results obtained by the authors in the context of clinical practice and to adequately highlight what contribution this study adds to the literature already existing on the topic and to future study perspectives.
Thank you for your suggestion. We have attempted to define the implications of our data for clinical practice in the discussion section. The key message we want to convey is that pathogenic BRCA1/2 large rearrangements may be missed if cancer centres only use Myriad's myChoice® HRD test to detect all BRCA1/2 pathogenic variants.